# CodeUnlearn: Amortized Zero-Shot Machine Unlearning in Language Models Using Discrete Concept

**YuXuan Wu**[1], **Bonaventure F. P. Dossou**[2,3], **Dianbo Liu**[1]

[1] National University of Singapore, Singapore, Singapore
[2] McGill University, Montreal, Canada
[3] Mila Quebec AI Institute, Montreal, Canada

## Abstract

Large Language Models (LLMs) offer extensive knowledge across various domains, but they may inadvertently memorize sensitive, unauthorized, or malicious data, such as personal information in the medical and financial sectors. Machine unlearning methods aim to remove specific information from models after training to address this. However, current approaches require additional model training or struggle to effectively erase particular data points and their associated context due to LLMs' complex, dense, and continuous nature. In this study, we propose a novel amortized unlearning approach using codebook features and Sparse Autoencoders (SAEs). By leveraging a bottleneck to decompose the activation space and regulate information flow, our method efficiently unlearns targeted information while preserving the model's performance on unrelated data. To the best of our knowledge, this is the first work that successfully enables unlearning specific topics with contextual relevance in an LLM, marking a significant step towards real-world applications of machine unlearning.

## 1 Introduction

Large language Models (LLMs) have been widely used in various applications, generating text responses that attempt to create the equivalent of human conversations OpenAI et al. (2024). These models leverage vast scientific literature to facilitate and accelerate interdisciplinary research Taylor et al. (2022) while drawing upon large datasets of human-generated content to provide professional advice. However, in many cases, such data is a double-edged sword. Including personal information or sensitive scientific knowledge can be beneficial or, conversely, harmful. For instance, Soice et al. (2023) discusses how LLMs, when used by non-experts, can enable the creation of biological agents, posing both potential benefits and significant risks.

In response to these concerns, machine unlearning has emerged as a promising research area focused on selectively removing specific data points or information from a trained model. This approach helps mitigate the misuse of sensitive data and addresses privacy concerns. Existing solutions, such as Sharded, Isolated, Sliced, and Aggregated (SISA) training Bourtoule et al. (2020), primarily involve partitioning the training data into disjoint shards and retraining models on these individual shards. Although effective in certain scenarios, these methods are often time-consuming, resource-intensive, and lack scalability when applied to large models like LLMs. Moreover, traditional approaches typically require specialized data structures or full retraining, making them impractical for dynamic or complex tasks.

Given these limitations, there is an increasing demand for zero-shot unlearning methods, which aim to remove specific information without retraining or specialized data structures. Unlike traditional unlearning techniques that rely on retraining portions of the model, zero-shot unlearning seeks to directly eliminate the influence of specific data points or pieces of information from the model's learned representation—without additional computational steps or parameter adjustments. More-

over, zero-shot unlearning is inherently more scalable, especially for large models like LLMs, as it avoids the inefficiencies associated with data partitioning and retraining.

Our approach builds upon using discrete representations as the latent space for unlearning. Discrete representations, generated through Vector Quantization (VQ) van den Oord et al. (2018), offer a natural structure for organizing the latent space to enable selective information removal. Discrete representations can be seen as a form of disentanglement, a concept rooted in classical research Bengio et al. (2014), which emphasizes learning representations that disentangle the various factors of variation in data. This allows for the separation of different explanatory sources within the data.

Additionally, Elhage et al. (2022) explores how neurons in models can represent multiple superposed features, introducing the concept of using dictionaries to disentangle these superpositions. Building on this notion, we propose employing discrete representations to disentangle the model's internal structure, thereby enabling selective unlearning. By tracking and modifying discrete codes within the latent space, we aim to achieve efficient and targeted removal of sensitive or unwanted information.

Our contributions are as follows:

- we propose a novel zero-shot unlearning method based on discrete latent representations.

- we demonstrate how Vector Quantization (VQ) can structure the latent space, facilitating the selective removal of information in an amortized manner.

- we extend our method beyond traditional machine unlearning techniques, primarily designed for classification tasks, to handle complex language tasks associated with language models, addressing a broader scope of applications.

- Our approach provides a baseline for unlearning in language models and validates the effectiveness of our method.

## 2 RELATED WORK

Machine unlearning methodologies have been developed to tackle the challenges of efficiently removing data from trained models. Among the early influential frameworks is the Sharded, Isolated, Sliced, and Aggregated (SISA) approach Bourtoule et al. (2020),which partitions data into independent shards. By retraining only the specific shards containing the data to be unlearned, SISA reduces the computational burden. Extensions of this approach include Ginart et al. (2019), which applies partitioning to linear models, and Brophy & Lowd (2021), which adapts it for random forests. Schelter et al. (2021) further extended the concept to decision trees, minimizing retraining through hierarchical partitioning. In the graph learning domain, Chen et al. (2022b) developed methods to forget specific nodes or edges, while Chen et al. (2022a) focused on removing sensitive user data from recommendation systems.

While these methods are effective for structured models, they struggle to scale to large, complex models like Language Models. Additionally, the retraining costs, though reduced, remain significant, and the reliance on specific architectures limits their generalizability to more dynamic tasks.

In a different direction, Kurmanji et al. (2023) introduced SCRUB, which treats the original model as a teacher and trains a student model to mimic it on retained data while 'forgetting' specific information. Warnecke et al. (2023) proposed unlearning entire groups of features and labels using influence functions, providing closed-form updates to model parameters for more efficient data removal.

Influence functions Guo et al. (2023); Sekhari et al. (2021); Mehta et al. (2022) also offer an alternative by measuring the effect of individual data points on a model's predictions and adjusting parameters accordingly, providing more direct methods for unlearning.

Recently, zero-shot unlearning methods have emerged, focusing on removing information without retraining, making them highly efficient for large models. Shah et al. (2024) introduced a method for editing model computations to 'forget' specific information. While this is effective for tasks like token classification, it may struggle with the more complex context and semantics in LLMs, underscoring the need for scalable, adaptable unlearning techniques tailored to these models.

## 3    METHODOLOGY

To address the challenges of zero-shot machine unlearning, we propose a novel approach that leverages *codebook features* to bottleneck latent representations within a language model, enabling the targeted unlearning of specific knowledge by altering related codebook embeddings. Initially introduced by Tamkin et al. (2023), codebook features efficiently compress the activation space of neural networks by introducing a sparse discrete bottleneck. This bottleneck can be further optimized to isolate the codes most relevant to specific topics in the input, offering deeper insight and control over the model's response and interpretation. By utilizing this discrete latent representation, we can more effectively identify and remove the specific information encoded in the codebook corresponding to the input's targeted knowledge.

The following section details our approach to employing *codebook features* to efficiently identify and unlearn specific areas of related information in a zero-shot manner. This process ensures that the model can no longer effectively handle prompts that contain the target information to unlearn.

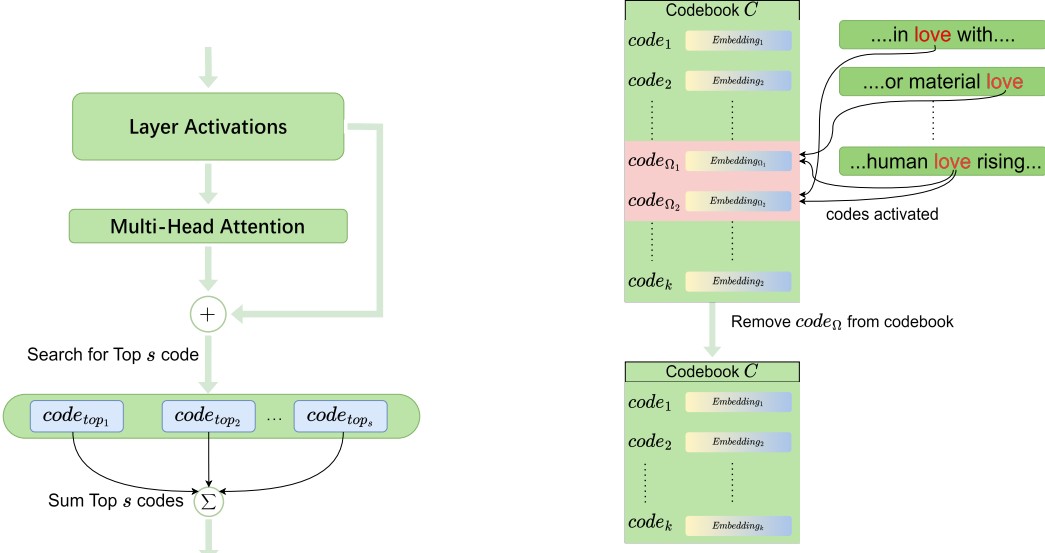

Figure 1: **CodeUnlearn**—Our Amortized Zero-Shot Machine Unlearning for Language Models. **Left**: Discrete latent bottlenecking in the transformer architecture. After applying the residual connection, the multi-head attention output is discretized using a discrete embedding vocabulary, referred to as the codebook. This approach prevents information leakage via the residual connection, ensuring that the codebook effectively regulates and interprets the network's behavior. **Right**: Zero-shot machine unlearning is achieved by removing the discrete codes in the codebook that correspond to the targeted information.

### 3.1    CODEBOOK FEATURES

The core concept behind employing codebook features is to transform the original activations from a hidden layer into a representation regulated by a codebook. Let $a \in \mathbb{R}^F$ represent the activation vector from a hidden layer, where $F$ denotes the dimensionality of the activations. We use a codebook $C = \{c_k\}_{k=1}^{K} \in \mathbb{R}^{K \times F}$, where $K$ represents the number of code vectors. The codebook offers a compressed, discrete representation of the original activations. To perform this transformation, we calculate the cosine similarity between the activation $a$ and each code vector $c_k$ in the codebook:

$$\text{cosineSim}(a, c_k) = \frac{a \cdot c_k}{\|a\|\|c_k\|}, \tag{1}$$

for each code vector $c_k$ in the codebook. We then identify the top $S$ (where $S \geq 1$) most similar code vectors corresponding to the activation $a$. The index set $\Omega$ of these top $S$ code vectors is defined as:

$$\Omega = \text{Top}_S\left(\{k \mid k \in \{1, \ldots, K\}, \text{cosineSim}(a, c_k)\}\right). \tag{2}$$

The output of the codebook transformation is given by:

$$\hat{a} = \sum_{k \in \Omega} c_k, \tag{3}$$

where $\Omega$ is the index set of the $S$ most similar code vectors, selected based on the highest cosine similarity scores. In the unlearning procedure, the activated codes corresponding to $a$ are identified as the targets for removal.

## 3.2 CODEBOOK SETTINGS

**Multiple Codebooks**    In prior work Tamkin et al. (2023), multiple codebooks were applied to each attention head, with the outputs concatenated across heads. Each attention head operates with its own codebook, selecting codes independently. The chosen codes from each head are then concatenated to produce the final output for that attention layer, effectively allowing the model to represent a broader set of features through the combination of different codebooks. Using multiple codebooks across attention heads can lead to a superposition effect, as described by Elhage et al. (2022). Superposition refers to the phenomenon where linear representations can encode more features than the dimensions, effectively allowing the neural network to simulate more extensive networks. In this case, combining multiple codebooks across attention heads allows for a significantly more comprehensive set of activations to be represented, even when using only the top $S = 1$ codebooks. However, tracking which individual codebooks contribute to specific activation patterns becomes challenging. Rather than relying on the output of a single codebook, the overall representation emerges from the combined outputs of all the codebooks.

**Single Codebook**    As shown in Section 3, to maintain interpretability, we focus on using a single codebook, positioning it after the multi-head attention layer and residual connection to prevent information leakage. However, in a single codebook setup, selecting only $S = 1$ leads to a significant drop in model performance, as a single codebook feature is insufficient to capture the complexity of the activation space. In Cai (2024), the author rigorously demonstrates that treating word vectors as mappings allows a finite vocabulary to achieve infinite approximation through composition. Based on this insight, we employ $S > 1$ in our approach. While this may slightly affect code discretization and information clarity, it strikes a balance between model performance and interpretability.

## 3.3 CODEBOOK WITH SPARSE AUTOENCODERS

Our goal is to decompose the activation space into sparse, interpretable features rather than reconstructing the original input. To accomplish this, we incorporate the Sparse Autoencoder (SAE) concept. The SAE applies a linear transformation encoder with a ReLU activation function to project the activations into a higher-dimensional space, effectively decomposing features. A linear transformation decoder is employed used to reconstruct the activations.

In line with the SAE structure, we introduce a linear transformation encoder with ReLU before the codebook and a linear transformation decoder after the codebook. This setup provides two significant benefits for machine unlearning:

- **Security through ReLU**: The ReLU activation function ensures that the extracted features are non-linear and sparse, making it more difficult to recover or reconstruct the original input from the features. This acts as a safeguard, reducing the likelihood of information leakage. By enforcing sparsity and non-linearity, ReLU provides greater control over feature representation, allowing us to obscure specific activations and protect data integrity during machine-unlearning processes.
- **Decentralization of Information**: Sparsity promotes the decentralization of encoded information, which helps isolate and unlearn specific patterns or features without disrupting the rest of the model. This targeted approach allows for more precise unlearning of sensitive or undesired information.

**Encoder**    The encoder is responsible for projecting the activation vector $a \in \mathbb{R}^d$ into a higher-dimensional space. This is achieved using a weight matrix $W_E \in \mathbb{R}^{d \times F}$ and a bias vector $b_E \in \mathbb{R}^d$.

A ReLU activation function follows the projection to introduce non-linearity:

$$h_{enc} = \text{ReLU}(W_{enc}a + b_{enc}). \tag{4}$$

**Codebook**   After encoding, the sparse representation $h_{enc}$ is transformed using the codebook. The cosine similarity between $h_{enc}$ and each code vector $c_k \in \{c_1, c_2, \ldots, c_K\}$ is calculated as:

$$\text{cosineSim}(h_{enc}, c_k) = \frac{h_{enc} \cdot c_k}{\|h_{enc}\|\|c_k\|}. \tag{5}$$

The top $S$ most similar code vectors are selected:

$$\Omega = \text{Top}_S\left(\{k \mid k \in \{1, \ldots, K\}, \text{cosineSim}(h_{enc}, c_k)\}\right). \tag{6}$$

The output of the codebook transformation is then:

$$\hat{h}_{enc} = \sum_{k \in \Omega} c_k. \tag{7}$$

**Decoder**   The decoder then maps $\hat{h}_{enc}$ back to the original activation space using a weight matrix $W_{dec} \in \mathbb{R}^{F \times d}$ and a bias vector $b_{dec} \in \mathbb{R}^F$:

$$\hat{a} = W_{dec}\hat{h}_{enc} + b_{dec}. \tag{8}$$

## 3.4   Training the Codebook

**Reconstruction Loss**   As with the Sparse Autoencoder (SAE) and codebook models, we utilize the Mean Squared Error (MSE) loss as the primary loss function. The MSE loss can be expressed as:

$$\mathcal{L}_{\text{MSE}} = \frac{1}{N} \sum_{i=1}^{N} \|a_i - \hat{a}_i\|_2^2, \tag{9}$$

where $N$ is the number of samples, $a_i$ is the original activation, and $\hat{a}_i$ is the reconstructed activation obtained from the decoder.

Additionally, to promote sparsity and enforce more distinct and sparse internal feature representations within each codebook vector, we introduce an $L_1$ penalty term on the codebook activations. This encourages the model to represent each code with sparser and more well-separated internal features. The overall loss function incorporating this sparsity constraint is defined as:

$$\mathcal{L}_{\text{Codebook}} = \frac{1}{N} \sum_{i=1}^{N} \|a_i - \hat{a}_i\|_2^2 + \lambda \sum_{k \in \Omega} \sum_{f=1}^{F} |c_k^f|, \tag{10}$$

where $\Omega$ represents the set of indices for the top $S$ most similar code vectors, $c_k$ refers to the $k$-th codebook vector, $F$ denotes the dimensionality of the code vectors, and $\lambda$ is a regularization coefficient that controls the strength of the $L_1$ penalty term. In our experiments, we set $\lambda$ to $1 \times 10^{-6}$ to balance sparsity with reconstruction accuracy.

**Joint Training for Machine Unlearning**   Both the SAE and codebook features are used to reconstruct the input $a$, but this presents a critical issue in the context of machine unlearning: one could easily remove the codebook layer, reverting the model to its original state, which negates the unlearning process. To address this, it is vital to ensure that the model is trained so that the downstream components are entirely dependent on the output of the codebook. At the same time, the upstream layers must learn to generate activations that conform to the codebook's representations. This joint training approach ensures that the entire model relies on the codebook's representation, making it harder to bypass or remove without degrading performance. The joint loss function for this training process is defined as:

$$\mathcal{L}_{\text{joint}} = \mathcal{L}_{\text{Codebook}} + \mathcal{L}_{\text{CE}}, \tag{11}$$

where $\mathcal{L}_{\text{Codebook}}$ refers to the reconstruction loss for the codebook, and $\mathcal{L}_{\text{CE}}$ represents the Cross-Entropy loss for the original language modeling or task-specific objective.

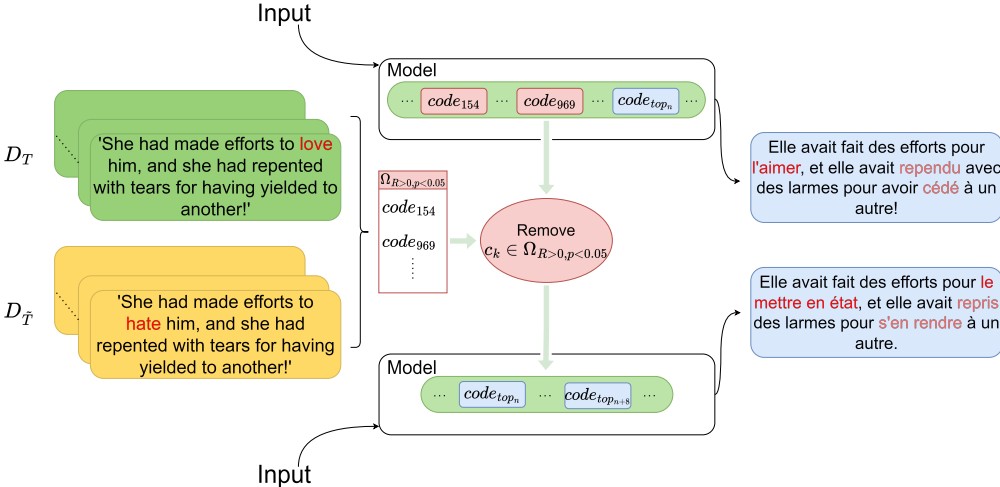

Figure 2: **Unlearning a Target Topic in a Language Model.** The zero-shot unlearning process begins by identifying codes enriched in data subsets with the target topic ($D_T$) as opposed to the subset without it ($D_{\tilde{T}}$). Codes with p-values less than 0.05 are removed from the codebook. After this removal, the model exhibits significantly decreased performance on target information inputs.

### 3.5 CODE RETRIEVAL

As shown in Figure 2, after training, the codebook encodes a set of representative codes $C = \{c_k\}_{k=1}^{K} \in \mathbb{R}^{K \times F}$ that are sparse and represent different features. To perform unlearning, we retrieve the codes activated for specific inputs and identify which codes are enriched for a particular topic. The model can effectively unlearn the associated information by deleting the corresponding enriched codes from the codebook. The key steps involve retrieving these relevant codes for each input and determining their relationship to the target topic.

Because of the nature of the attention mechanism, the activation of these codes also depends on the surrounding context. This means we are not just identifying individual words that activate specific codes but retrieving codes that represent the broader topic within the input context. To unlearn a specific topic $T$, consider a dataset $D_T$ with samples related to topic $T$, alongside with the remaining irrelevant data set $D_R$. We create a control dataset $D_{\tilde{T}}$ by replacing words associated with $T$ in $D_T$ with unrelated words, ensuring the context remains consistent. By comparing the code activations between $D_T$ and $D_{\tilde{T}}$, we can identify and search for the codes linked to topic $T$.

For each code $c_k$ activated in the dataset, we compute its frequency in both datasets by considering the top $S'$ activated codes:

$$f_k(D_T) = \frac{1}{N_T} \sum_{i=1}^{N_T} \mathbb{I}(k \in \Omega_T(a_i)), \tag{12}$$

$$f_k(D_{\tilde{T}}) = \frac{1}{N_{\tilde{T}}} \sum_{j=1}^{N_{\tilde{T}}} \mathbb{I}(k \in \Omega_{\tilde{T}}(a_j)), \tag{13}$$

where $\Omega_T(a_i)$ represents the set of indices of the top $S'$ activated codes for activation $a_i$ in dataset $D_T$, and $\Omega_{\tilde{T}}(a_j)$ is similarly defined for $D_{\tilde{T}}$. $N_T$ and $N_{\tilde{T}}$ denote the sample sizes of $D_T$ and $D_{\tilde{T}}$, respectively. $\mathbb{I}$ is the indicator function that checks whether code $k$ is in the set of activated codes. The hyperparameter $S'$ controls the number of top activated codes considered, thereby influencing the number of codes to be removed.

To quantify the enrichment of code $c_k$ for topic $T$, we use the following formula:

$$\text{R}(c_k, T) = \log_2 \left( \frac{f_k(D_T) + \epsilon}{f_k(D_{\tilde{T}}) + \epsilon} \right), \tag{14}$$

where $\epsilon$ is a small constant added to avoid division by zero. When $R(c_k, T)$ is positive, it indicates that the code $c_k$ is enriched in dataset $D_T$ relative to $D_{\tilde{T}}$. However, if the frequency of $c_k$ in $D_{\tilde{T}}$

is zero and its frequency in $D_T$ is very low, such codes should not be removed, as they are likely accidental activations. Removing these codes could lead to unintended side effects, as they may not be strongly related to the topic $T$ despite being present in the dataset.

Therefore, we used a chi-squared test to calculate the p-value of $R(c_k, T)$ to determine if the code $c_k$ is enriched for topic $T$. For those codes with p-values smaller than 0.05, we regard them as enriched codes in $D_T$ and remove them from the codebook. We define the set of enriched codes as $\Omega_{R>0,p<0.05} = \{c_k \mid R(c_k, T) > 0 \text{ and } p \leq 0.05\}$.

## 3.6 METRICS

In our work, we not solely assess the absolute drop in performance within the topic or non-topic datasets but also need to compare the relative decline between them. Instead, to fairly compare the models and the datasets, we used normalized percentage improvement to evaluate the performance of the unlearning procedure. The performance improvement percentage is set to 0 for the zero-shot model and 1 for the codebook model, which is the upper bound. In contrast, the performance drop percentage is set to 1 for the zero-shot model and 0 for the codebook model. We use four evaluation metrics to assess the effectiveness of the unlearning procedure and the overall quality of the remaining information in the output. These metrics include: We use four evaluation metrics to assess the impact of the unlearning procedure on translation quality and semantic preservation: BLEUPapineni et al. (2002), METEORBanerjee & Lavie (2005), BERTScoreZhang et al. (2020), and Bart-ScoreYuan et al. (2021). BLEU offers a general accuracy measure, and METEOR builds on BLEU by considering synonymy and word order, often providing a more sensitive quality assessment. BERTScore leverages contextual embeddings to evaluate semantic similarity, crucial for detecting whether unlearning procedures change the sentence's meaning. Bart-Score evaluates fluency and informativeness using pre-trained BART models, with scores reflecting log-likelihood, so close to zero indicates better quality. BERTScore and Bart-Score offer insight into more subtle changes, and percentage change trends are prioritized for a comprehensive analysis.

Table 1: Examples of unlearning on topic '*love*'

|  | Content |
|---|---|
| **English** | She had made efforts to love him, and she had repented with tears for having yielded to another! |
| **Ground Truth** | Elle avait fait des efforts pour l'aimer, et elle s'était repentie en pleurant d'avoir cédé à un autre. |
| **Codebook Model** | Elle avait fait des efforts pour l'aimer, et elle avait repris des larmes pour avoir renoncé à un autre! |
| $S' = 8$, delete 16 codes | Elle avait fait des efforts pour l'aimer, et elle avait repris des larmes pour l'avoir acquitté d'un autre! |
| $S' = 24$, delete 52 codes | Elle avait fait des efforts pour le recevoir, et elle avaitrepris des larmes pour avoir renoncé à un autre. |
| $S' = 72$, delete 133 codes | Elle avait fait des efforts pour le mettre en état, et elle avait repris des larmes pour s'en rendre à un autre. |

## 4 EXPERIMENTS AND RESULTS

We applied the codebook features combined with SAE on a large language model(LLM) and trained it on tasks that exhibit clear distinctions between correct and incorrect answers. After training, we unlearned the model on several specific topics to measure the degradation in performance on the unlearned issues while ensuring minimal impact on the other topics. An example of the unlearning effect on the topic of '*love*' is shown in Table 1. The results illustrate that as more codes related to the target topic were deleted, the model's translation became less accurate in representing the original meaning. For instance:

The translation introduces minor inaccuracies in the case of $S' = 8$ (16 codes deleted). As the number of deleted codes increases to $S' = 72$ (133 codes deleted), the translation significantly deviates

from the original meaning, showing the model's inability to maintain accuracy on the target topic. This demonstrates that the model successfully forgets the '*love*' concept and the wrong meaning can even interfere with the rest of the sentences.

Table 2: **Unlearning Results for Different Topics**

| Topic(N) | Dataset | Score (Normalized Improvement Drop(%)) | | | |
|---|---|---|---|---|---|
| | | $BLEU\downarrow$ | $METEOR\downarrow$ | $BERT-P\downarrow$ | $BART\downarrow$ |
| **Love(207)** | $D'_T$ | 0.16 *(-112.52)* | 0.39 *(-117.76)* | 0.80 *(-118.88)* | -4.80 *(-143.96)* |
| | $D_R$ | 0.18 *(-37.80)* | 0.42 *(-57.82)* | 0.81 *(-58.25)* | -5.71 *(-35.06)* |
| **Julien(255)** | $D'_T$ | 0.19 *(-113.12)* | 0.42 *(-138.47)* | 0.80 *(-134.60)* | -5.15 *(-164.68)* |
| | $D_R$ | 0.16 *(-65.70)* | 0.39 *(-64.38)* | 0.80 *(-94.63)* | -6.10 *(-94.60)* |
| **Captain(137)** | $D'_T$ | 0.20 *(-72.10)* | 0.47 *(-140.71)* | 0.83 *(-84.44)* | -5.16 *(-87.90)* |
| | $D_R$ | 0.19 *(-9.72)* | 0.44 *(-9.04)* | 0.82 *(-9.66)* | -5.97 *(-0.53)* |
| **Poor(151)** | $D'_T$ | 0.18 *(-70.61)* | 0.43 *(-70.78)* | 0.81 *(-60.84)* | -5.03 *(-79.81)* |
| | $D_R$ | 0.20 *(-26.64)* | 0.47 *(-12.48)* | 0.83 *(-14.20)* | -5.81 *(-36.01)* |
| **Wish(217)** | $D'_T$ | 0.15 *(-144.83)* | 0.33 *(-249.51)* | 0.78 *(-182.02)* | -4.95 *(-309.34)* |
| | $D_R$ | 0.16 *(-87.65)* | 0.39 *(-94.51)* | 0.81 *(-74.16)* | -6.02 *(-133.35)* |
| **White(179)** | $D'_T$ | 0.12 *(-157.45)* | 0.38 *(-218.04)* | 0.80 *(-403.04)* | -4.85 *(-119.99)* |
| | $D_R$ | 0.16 *(-10.09)* | 0.49 *(-22.99)* | 0.83 *(-47.65)* | -6.12 *(-27.15)* |
| **Black(190)** | $D'_T$ | 0.16 *(-85.16)* | 0.40 *(-138.04)* | 0.80 *(-115.56)* | -4.70 *(-62.91)* |
| | $D_R$ | 0.19 *(-16.12)* | 0.47 *(-2.15)* | 0.83 *(-3.01)* | -5.78 *(-97.36)* |

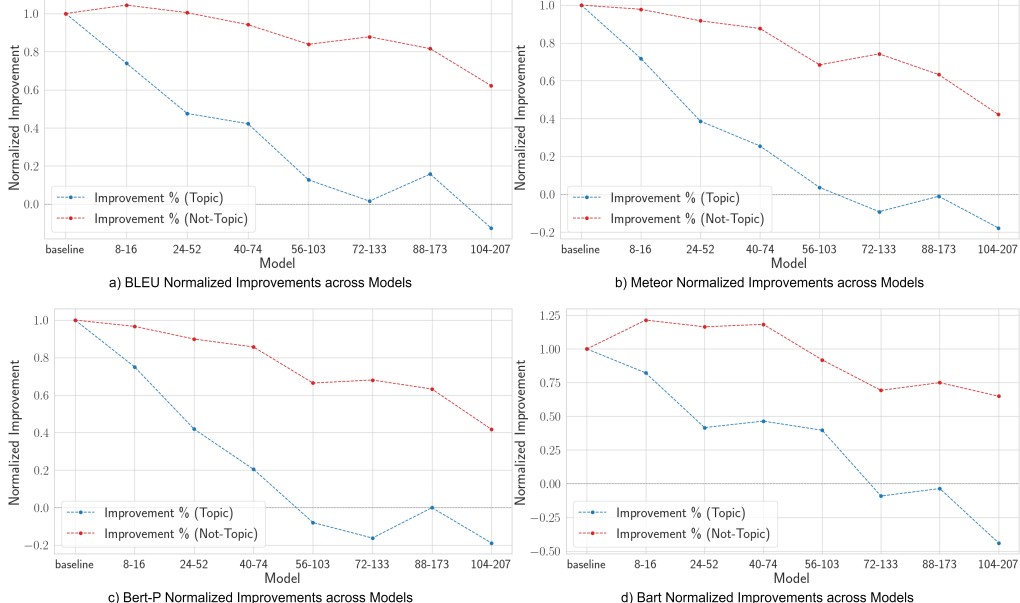

Figure 3: **Performance Drop after Unlearning on the Topic '*Love*'.** Performance Drop after Unlearning on the Topic '*Love*'. The X-axis shows the model variations, with the first column as the original model. Columns 2 to 8 represent increasing levels of unlearning, with the number indicating the top $S$ codes used and removed. The Y-axis represents the percentage change in various metrics compared to the original model. As more codes are deleted, the model's performance on the target topic declines rapidly, while performance on non-topic content remains more stable.

**Dataset Building** The dataset comprises three parts: (1) training, (2) validation, and (3) test datasets. The training dataset is used for both training and unlearning, while the validation and test datasets assess the performance of the unlearned model. For the unlearning procedure, we filtered prompts containing target words, sampling 500 instances for $D_T$ and then generated $D_{\tilde{T}}$. All relevant prompts from the test and validation datasets were used to create the dataset $D'_T$, while irrelevant prompts were used to construct the dataset $D_R$ for evaluation. We trained a T5-small model Raffel et al. (2023) with codebook features on the opus_books/en-fr dataset. A codebook with 25k codes and 512 dimensions was applied at the third layer of the encoder, as this layer likely captures more abstract, high-level features, ideal for our approach Templeton et al. (2024).

After training, we identified specific topics within the training dataset and performed the unlearning procedure. We tested seven values for $S'$ ranging from $8(1 \times S)$ to $104(13 \times S)$, each resulting in a different number of deleted codes. This led to a deletion of approximately 0.064% to 0.828% of the total codes in the codebook.

As shown in Figure 3, as the number of searched and deleted codes increases, the performance on the topic deteriorates rapidly. Although performance on non-topic deteriorates simultaneously, it is far better than the topic. For instance, in the case of the '*love*' topic, when $S' = 104(13 \times S)$, which corresponds to searching for the top 104 most similar codes in the codebook for each activation, about 0.828% of the codes were deleted. The improvement score for the target topic became negative, which means the unlearned model is worse than the zero-shot model. In contrast, the model's performance on non-topic is far better than the topic, demonstrating effective unlearning of the specific target while maintaining reasonable performance on unrelated information.

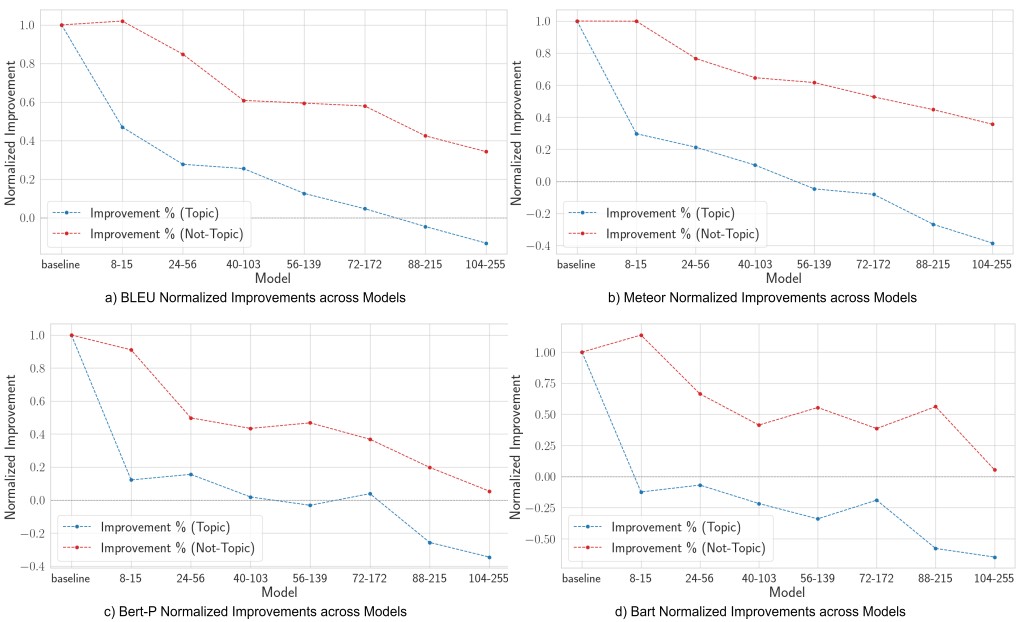

Figure 4: **Performance Drop after Unlearning on the Topic '*Julien*'.** Similar to the '*love*' topic, we tested the unlearning procedure on the name '*Julien*'.

Beyond conceptual topics like '*love*,' we also applied the unlearning procedure to the frequently occurring name '*Julien*' in the dataset. Names carry specific semantic significance in language models, much like critical topics, making '*Julien*' an ideal test case to assess the method's effectiveness in removing personal information, such as names, while preserving performance on unrelated content. As shown in Figure 4, the unlearning process led to a noticeable performance decline for '*Julien*' as the number of removed codes increased. Similar to the '*love*' topic, the model's performance on non-target content remained relatively stable. This further illustrates the versatility of the proposed approach in effectively unlearning targeted information, whether it is conceptual (like '*love*') or personal (like '*Julien*'), while maintaining accuracy on non-topic content. Following unlearning, the model attempts to rely on other similar codes; however, the meanings of these codes are signif-

icantly different. As a result, the unlearned target topic interferes, hindering the model's ability to comprehend the entire sentence fully.

In addition to the '*love*' and '*Julien*' topics, we performed unlearning on several other topics such as '*Captain*,' '*Poor*,' '*Wish*,' '*White*,' and '*Black*.' Table 2, shows the performance degradation across various topics after applying the unlearning procedure, with the number of deleted codes indicated in parentheses next to each topic. The values represent actual scores and the normalized improvement drop in performance, calculated relative to the zero-shot and baseline models before unlearning. A negative value indicates a performance decline. As $S'$ increases (for instance, $S' = 13 \times 8$ here), the performance gap between $D'_T$ and $D_R$ widens, demonstrating effective unlearning of the target topic with minimal impact on irrelevant information. To further assess the unlearning performance,

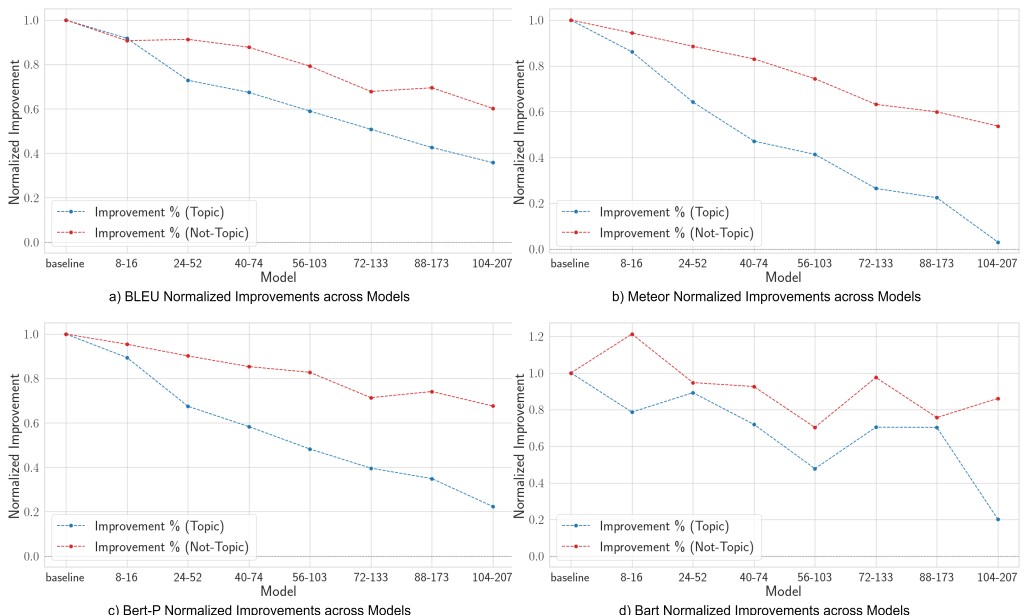

Figure 5: **Metrics after unlearning topic '*love*' and test on '*like*'**, The model unlearned the '*love*' topic but also deteriorated the performance on the '*like*' topic, which suggests that the unlearning procedure removes not only the specific target information but also the relevant context.

we also evaluate the synonymy of the target word, such as '*like*' in place of '*love*' shown in Figure 5. Ideally, the model's performance on the '*like*' topic should also worsen, suggesting that the unlearning procedure removes the specific target information and the broader context related to that concept. Our approach diverges from traditional data-point-unlearning tasks by removing the codes close to the activation space, which is essential in unlearning conceptual or contextual knowledge rather than isolated instances.

## 5 CONCLUSION

In this work, we introduced CodeUnlearn, a novel framework for zero-shot machine unlearning in Large Language Models (LLMs). Leveraging codebook features and Sparse Autoencoders (SAEs), we devised a method that effectively isolates and removes specific knowledge, ensuring that the targeted data and its contextual associations are erased from the model. Unlike previous methods, which required retraining or were limited to classification tasks, CodeUnlearn operates amortized and zero-shot, providing an efficient and scalable solution for unlearning in complex, generative models like LLMs. Our approach uses a discrete concept representation to regulate the flow of information in a language model, enabling the unlearning of specific topics while preserving overall model performance on unrelated tasks. The results show that CodeUnlearn successfully mitigates the model's ability to reproduce the unlearned information without requiring additional training, achieving substantial unlearning effectiveness and maintaining interpretability.

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

## A  TRAINING AND OPTIMIZATION DETAILS

This section provides additional details on the training and optimization of the Sparse Autoencoder (SAE) used in CodeUnlearn.

After the SAE encoder layer, we apply layer normalization to stabilize training and improve convergence. The dimensionality of the SAE is set to match both the codebook and input dimensions, which is 512.

For the initialization of the SAE encoder layer, we use Kaiming uniform initialization He et al. (2015), which is well-suited for layers with ReLU activation. This method helps maintain the proper scale of the weights, preventing issues such as vanishing gradients. Additionally, since the codebook can be regarded as an activation layer, Kaiming initialization ensures that the input distributions to the codebook remain stable, facilitating efficient learning and representation of sparse features within the SAE.

To promote sparsity in the activations, we introduce an $l_1$ loss with a lambda parameter set to $1 \times 10^{-6}$. This ensures that the network learns sparse representations, which are crucial for enhancing the interpretability and control required for the unlearning process.

Codebook size is 25k and the dimensionality is 512, we use top 8 codes to represent the input.

## B  SEARCHING AND RETRIEVAL PROCEDURE

### B.1  DATA BUILDING

**Selection of $D_T$:**  We sampled 500 prompts containing the target words from the validation and test dataset.The validated prompt never participates in the training and unlearning phases. We first analyze word frequencies across the entire dataset to construct the target dataset $D_T$. We select words with frequencies between 500 and 700. Words that are too frequent tend to be overly familiar and lack specificity, while those that are too infrequent may not provide meaningful insights. We focus on words in the 500-700 frequency range, such as 'love,' which are practically meaningful and suitable for testing the unlearning process.During validation, we created $D_T'$ by selecting topic-specific prompt components from the test and validation sets, and we sampled an equal number of instances from the remaining irrelevant dataset to construct $D_R$.

**Generation of $D_{\tilde{T}}$:**  For the control dataset $D_{\tilde{T}}$, we replace the target words in $D_T$ with common non-synonyms of the same part of speech. The replacement words are selected based on word frequencies reported by Norvig (2009). For instance, for names, we randomly generate other names to replace the original ones. This ensures that $D_{\tilde{T}}$ maintains the same contextual structure as $D_T$, allowing us to focus on how effectively the unlearning procedure targets specific information.

## B.2  SEARCH AND RETRIEVAL OF CODES

For code search and retrieval, we disable sampling by setting the temperature to 0 at all stages, ensuring deterministic behavior in code activation selection.

Table 3: Runtime Mean and Standard Deviation for Different $S'$

| $S'$ | Runtime Mean (s) | Runtime Std (s) |
|---|---|---|
| 8 | 473.66 | 264.58 |
| 24 | 376.98 | 238.66 |
| 40 | 212.35 | 240.88 |
| 56 | 211.23 | 438.63 |
| 72 | 211.14 | 479.11 |
| 88 | 214.12 | 434.29 |
| 104 | 215.37 | 526.23 |

As shown in Table 3, the runtime varies significantly due to the different lengths of the prompts. Despite this fluctuation, it can be observed that the average search time for the top 500 samples is approximately 10 minutes, indicating an efficient unlearning process.

## C  EXAMPLES OF UNLEARNING

Table 4: Examples of unlearning on the topic '*Julien*'

|  | Content |
|---|---|
| **English** | Without being the least bit in the world intimidated, Julien resumed his narrative. |
| **Ground Truth** | Sans être le moins du monde intimidé, Julien reprit sa narration. |
| **Codebook Model** | Sans être le moindre obstacle du monde, Julien reprit son récit. |
| $S' = 8$, delete 16 codes | Sans être le moindre obstacle du monde, je reprit son récit. |
| $S' = 24$, delete 52 codes | Sans être le moindre objet du monde attaqué, le temps lui reprit son récit. |
| $S' = 72$, delete 133 codes | Sans être le moindre obstacle du monde, M. Rochester reprit son récit. |

As shown in Table 4, by $S' = 24$, deleting 52 codes already leads to a significant performance drop. The name '*Julien*' is no longer recognized after code deletion, and the model attempts to fill this gap with unrelated words. This behavior interferes with the model's understanding of the context, as it tries to substitute Julien's code with alternatives, making it impossible to restore the correct information. The model provides incorrect substitutions, rather than leaving the slot vacant for further inference.

In Table 5, we observe that the model's performance on unrelated content, like the '*Notre—Dame*' topic, remains relatively stable even after unlearning the '*Julien*' topic. Only minor perturbations occur at higher code deletions (e.g., $S' = 72$), but the overall sentence retains its meaning, demonstrating the model's resilience on non-target content. The resulting change, which involves a preposition shift, has a negligible effect on the overall meaning of the sentence, further confirming that the unlearning process effectively targets only the specified concept without broadly disrupting unrelated text generation.

## D  FUTURE WORK

While CodeUnlearn has demonstrated its effectiveness in unlearning specific topics in LLMs, several areas remain for further exploration:

Table 5: Non-topic samples after unlearning on the topic '*Julien*'

|  | **Content** |
|---|---|
| **English** | In fact, within the bounds of Notre—Dame, the condemned girl could not be touched. |
| **Ground Truth** | En effet, dans l'enceinte de Notre—Dame, la condamnée était inviolable. |
| **Codebook Model** | En effet, dans les limites de Notre—Dame, la condamnée ne pouvait être touchée. |
| $S' = 8$, delete 16 codes | En effet, dans les limites de Notre—Dame, la condamnée ne pouvait être touchée. |
| $S' = 24$, delete 52 codes | En effet, dans les limites de Notre—Dame, la condamnée ne pouvait être touchée. |
| $S' = 72$, delete 133 codes | En effet, au milieu des limites de Notre—Dame, la condamnée ne pouvait être touchée. |

- **Enhanced Code Retrieval with Minimal Impact on Unrelated Information**: Improving the accuracy of identifying target codes can lead to more precise unlearning with reduced unintended consequences on irrelevant information. Future work could focus on refining the search and retrieval process to ensure that unlearning specific knowledge has minimal impact on the model's overall performance and generalization capabilities.

- **Decentralized Code Representation**: One goal is to decentralize further the information encoded in the codebook to ensure that unlearning-specific features have an even more localized impact on the model's behavior. This could lead to finer control over the granularity of the unlearning process.

- **Expanding to Other Tasks and Architectures**: While our method has been validated on language models, expanding **CodeUnlearn** to tasks like classification and extending it to other model architectures (e.g., transformers beyond T5) will further enhance its applicability across domains.

## E  FURTHER DETAILS ON TRADITIONAL UNLEARNING METHODS

In this appendix, we delve deeper into some of the traditional machine unlearning methods, expanding on the frameworks and strategies discussed in the related work section.

**SISA (Sharded, Isolated, Sliced, and Aggregated) Approach**    The Sharded, Isolated, Sliced, and Aggregated (SISA) approach Bourtoule et al. (2020) partitions the training data into independent shards, each used to train isolated models or sub-models. When a specific data point needs to be unlearned, only the relevant shard containing that data is retrained. This approach is designed to improve computational efficiency by reducing the need for full model retraining.

While SISA is highly efficient compared to retraining the entire model, the framework introduces certain challenges. The isolated training of each shard can result in a lack of information integration across different shards, potentially leading to generalization issues. In large language models (LLMs), where complex interdependencies between tokens are crucial for performance, the isolated shard approach can cause degradation in performance. Moreover, as the size of the dataset grows, the retraining costs, even within individual shards, remain significant, making SISA less practical for large-scale LLMs.

**Extensions to SISA: DaRE, HedgeCut, and ARCANE**    Other methods such as DaRE Brophy & Lowd (2021) and HedgeCut Schelter et al. (2021) extend SISA's principles to tree-based algorithms. These approaches focus on partitioning the decision tree structure to ensure that only specific branches or paths are retrained during unlearning. DaRE adapts the SISA framework for random forests, while HedgeCut applies it to hierarchical decision trees, offering more flexibility across different model architectures.

ARCANE Yan et al. (2022) represents another evolution of the SISA framework by optimizing retraining costs through class-based partitioning. In ARCANE, the dataset is divided into class-specific subsets, minimizing the impact of unlearning by only requiring retraining for the class in question. This strategy enhances efficiency by limiting the scope of retraining, but it still necessitates retraining, which can become a bottleneck, especially for high-dimensional and large-scale datasets.

**Limitations of SISA and Its Variants in Complex Models**  Despite the advancements made by SISA and its extensions, these methods rely heavily on specific model architectures and data structures, making them less suitable for complex and unstructured environments like LLMs. In large language models, the intricate dependencies between tokens mean that partitioning the data into isolated shards or classes may not capture the full complexity of the model's learned representations.

The isolated training across shards can also lead to issues with model generalization, as each shard is trained independently. This becomes particularly problematic when the model needs to generalize to unseen data. The lack of integration between shards can cause performance degradation, particularly in tasks requiring high-level contextual understanding, such as those found in LLMs. Moreover, although SISA limits retraining to individual shards, the computational burden remains substantial for large-scale datasets, making the approach less scalable for real-world deployment in LLMs.

**Influence Functions for Unlearning**  An alternative to retraining-based methods is the use of influence functions, which estimate the impact of a data point on the model's learned parameters Guo et al. (2023); Sekhari et al. (2021); Mehta et al. (2022). Influence functions allow the model to reverse the effects of specific data points without needing full retraining. By calculating the gradient of the loss function with respect to the training points, influence functions can adjust the model's parameters to 'forget' the data.

However, while influence functions are efficient for simple models like linear classifiers or small neural networks, they struggle with the complexity and non-linearity of deep architectures like LLMs. The dense and interconnected structure of LLMs makes it difficult to isolate the effect of individual data points without affecting the model's overall performance. This limitation restricts the scalability of influence functions in unlearning tasks within complex models.

**Re-optimization After Unlearning**  A novel approach to selective forgetting, based on re-optimization, was proposed by Golatkar et al. (2019), who introduced an optimal quadratic scrubbing algorithm designed to achieve selective forgetting in deep networks. Selective forgetting is defined as the process of modifying network weights using a scrubbing function $S(w)$, such that the weight distribution becomes indistinguishable from that of a network never trained on the forgotten data. This is quantitatively measured through the Kullback-Leibler (KL) divergence. If the KL divergence between the post-scrubbing weight distribution and the weight distribution of a network that has never encountered the forgotten data approaches zero, it indicates complete forgetting. This method ensures that the network 'forgets' specific information without necessitating full retraining, and instead re-optimizes the network's weights to achieve a distributional equivalence.

However, one of the key limitations of this approach is its computational complexity. While the scrubbing process avoids full retraining, re-optimization still involves significant computational overhead, especially for large-scale models like LLMs. Additionally, achieving true distributional equivalence is highly challenging in practice, particularly when the network is fine-tuned on multiple tasks or trained on diverse datasets. This often leads to incomplete forgetting, as small traces of the forgotten data may still influence the network's behavior.

Building on the idea of re-optimization, Shibata et al. (2021) introduced the Learning with Selective Forgetting (LSF) framework, which aims to selectively forget specific classes in a lifelong learning setting. LSF employs a multi-component loss function that balances classification accuracy, mnemonic embedding, selective forgetting, and regularization to prevent catastrophic forgetting of non-target classes. This method, though promising, suffers from scalability issues when applied to larger datasets or more complex models. The reliance on class-level removal also limits its applicability to scenarios where granular, instance-level forgetting is required, making it less adaptable to tasks beyond classification, such as generative language models.

Furthermore, both approaches struggle with model interpretability and traceability post-unlearning. As the network weights are continuously re-optimized, it becomes difficult to verify the extent of

forgetting or to ensure that no residual influence from the forgotten data remains. The lack of guarantees about complete data removal can be a significant concern in privacy-sensitive applications, where even small data remnants could pose risks. This calls for more transparent and auditable unlearning processes, particularly in contexts involving sensitive personal or confidential information.

**Re-optimization After Unlearning**  Re-optimization-based approaches to selective forgetting, such as the quadratic scrubbing algorithm proposed by Golatkar et al. (2019), aim to adjust a model's weights so that the distribution resembles one that has never been exposed to the forgotten data. This is measured using Kullback-Leibler (KL) divergence, with the goal of reducing it to near zero, indicating complete forgetting without full retraining. While effective, this method is computationally expensive, especially for large models like LLMs, and achieving perfect distributional equivalence is difficult, often leaving residual traces of the forgotten data.

The Learning with Selective Forgetting (LSF) framework introduced by Shibata et al. (2021) enhances this by incorporating a loss function that balances accuracy, mnemonic embedding, selective forgetting, and regularization to remove specific classes in lifelong learning. However, both methods face scalability challenges with large datasets and struggle with more granular, instance-level forgetting required in complex tasks like language generation.

Moreover, these approaches lack transparency and traceability, making it difficult to verify whether forgetting has been truly achieved. This is particularly problematic in privacy-sensitive contexts, where even minor data remnants can pose significant risks. Thus, re-optimization methods, while promising, require further refinement to handle large-scale models and ensure complete, verifiable unlearning.

## F  FURTHER DETAILS ON VECTOR QUANTIZATION METHODS

A promising direction to address these challenges lies in Vector Quantization (VQ) and Sparse Coding, which provide a natural framework for disentangling information encoded in models, offering deeper insights into model interpretability Elad (2010). Numerous studies have demonstrated the effectiveness of sparse vectors in discovering underlying sparse structures, significantly improving interpretability.

For example, Arora et al. (2018) showed how sparse coding can reveal the linear algebraic structure of word embeddings, enhancing their interpretability. Similarly, Olshausen & Field (1996), along with Donoho & Elad (2003), explored how sparse coding in visual systems identifies the most relevant features, underscoring the potential of sparse representations for revealing meaningful features in complex models.

Expanding on these ideas, Shah et al. (2023) proposed a Discrete Key-Value Bottleneck (DKVB) model that leverages sparse representations, freezing key-value pairs to prevent gradient propaga-

tion and enabling unlearning without retraining. While effective for classification tasks, the DKVB model faces challenges when applied to large language models (LLMs) due to the more intricate relationships between tokens and context, highlighting the need for unlearning methods better suited to the complexity of LLMs.

More recently, Elhage et al. (2022) demonstrated how sparse coding can extract and disentangle superpositions in toy models, providing valuable insights into the structure of neural networks. By applying sparse coding techniques, Elhage et al. (2022) were able to disentangle these superpositions, offering a clearer understanding of the complex behaviors observed in deep neural networks.

Building on these advancements, Sparse Autoencoders (SAE) further enhance model interpretability by decomposing activation spaces into distinct, sparse components Templeton et al. (2024). SAEs allow models to identify specific features where information is encoded, making it easier to selectively remove or modify individual components during the unlearning process. By leveraging the sparsity and disentanglement properties of VQ and SAE, it is possible to develop unlearning methods that are scalable, efficient, and interpretable, offering a robust alternative to techniques that rely on retraining or complex data partitioning.

