# OpenReview forum: "CodeUnlearn: Amortized Zero-Shot Machine Unlearning in Language Models Using Discrete Concept"
_NeurIPS.cc/2024/Workshop/SafeGenAi — SafeGenAi Poster_

### Official Review · Reviewer_i4Ph · 2024-10-09
**Review of Submission 92**

**Rating:** 7
**Confidence:** 3

**Review:**

This paper proposes a novel algorithm for zero-shot machine unlearning. The proposed new framework, **CodeUnlearn**, maintain a codebook of the model's output. To remove the concept, it removes the unlearn concepts in the codebook. I think the framework is reasonable, and the paper is well written. Meanwhile, the topic is relevant to the workshop.

One weakness is that is paper does not experiment on the normal dataset, which makes the effectiveness of the proposed algorithm remains unclear. It should do one experiment on testing model's performance on standard NLP benchmark, before and after the unlearning process. Another weakness is that the memory size of the codebook is unclear, which makes me doubt the scalability of this method.

---

### Official Review · Reviewer_GRv1 · 2024-10-09
**This paper exceeds the page limit set by the submission guidelines.**

**Rating:** 1
**Confidence:** 5

**Review:**

I must point out that this paper exceeds the page limit set by the submission guidelines.
I recommend revising the paper to adhere to the required length

---

### Official Review · Reviewer_evCj · 2024-10-09
**Ok to accept**

**Rating:** 7
**Confidence:** 3

**Review:**

This paper presents CodeUnlearn, a novel zero-shot unlearning method for language models that utilizes discrete codebook features and Sparse Autoencoders (SAEs) to remove specific knowledge from a model without requiring retraining. This approach addresses the key challenges faced by traditional machine unlearning techniques, such as high computational costs, limited scalability, and reduced effectiveness for complex models like LLMs.

## Pros
1. Novelty and Significance: The use of discrete codebooks for unlearning is an innovative approach that has not been applied to LLMs before. By bottlenecking the activation space and isolating discrete codes, this method achieves targeted unlearning without harming unrelated tasks, which is crucial for real-world applications such as data privacy and compliance.
2. Scalability: The method is amortized and zero-shot, making it highly scalable for large models, overcoming the limitations of retraining-based approaches like SISA and SCRUB.
3. Comprehensive Methodology: The paper provides a well-structured explanation of the methodology, with details on how the codebook and SAE are trained and used to selectively forget information. This is backed by a mathematical formulation, which makes the contribution clear and verifiable.
4. Experimental Validation: The experiments clearly demonstrate the effectiveness of CodeUnlearn on a variety of topics (e.g., “love”, “Julien”). The metrics used (BLEU, METEOR, BERTScore) are appropriate and illustrate a consistent drop in performance on unlearned topics without significantly affecting other topics. The results provide convincing evidence of the method’s practical utility in unlearning.
5. Contextual Unlearning: The approach effectively handles conceptual unlearning, ensuring that unlearning does not only target specific data points but also the contextual relevance of the knowledge, which is crucial for tasks involving LLMs.

## Cons
1. Granularity of Unlearning: While the unlearning of topics like “love” and “Julien” is successful, the paper does not sufficiently explore how granular the unlearning process can be. More discussion on how the method handles nuanced or less clearly defined topics would improve the work’s generalizability.
2. Impact on Synonyms and Related Concepts: The experiment on the “love” topic showed that unlearning also affected the synonym “like”. While this might indicate successful removal of the broader conceptual space, the potential impact on related concepts could undermine some use cases. More detailed analysis is needed on how the method balances precision and recall of unlearned knowledge.
3. Computational Efficiency: While the paper claims that the method is scalable, it lacks detailed analysis of the computational overhead involved in training the codebook and conducting the unlearning process for very large models. A direct comparison with state-of-the-art methods in terms of runtime and resource requirements would be valuable.
4. Generalization to Other Architectures: Although the method is demonstrated on a specific LLM architecture, its applicability to a broader range of models remains unclear. Expanding on how the method can be generalized to architectures beyond T5 would increase its impact and relevance.

---

### Official Review · Reviewer_RpF2 · 2024-10-10
**Review of "CodeUnlearn: Amortized Zero-Shot Machine Unlearning in Language Models Using Discrete Concept"**

**Rating:** 6
**Confidence:** 4

**Review:**

The paper addresses an important and timely issue: how to effectively remove specific information from trained LLMs without the need for retraining, which is crucial for privacy and compliance reasons.
## Quality
* Methodology: The proposed approach is innovative, combining codebook features with SAEs to create a bottleneck that regulates information flow. This is a clever way to enable zero-shot unlearning.
* Experiments: The authors conduct experiments demonstrating the effectiveness of their method in unlearning specific topics while retaining performance on other data.
* Results: Quantitative metrics like BLEU, METEOR, and BERTScore are used to evaluate the unlearning effect, showing significant degradation in performance on the unlearned topics.
## Clarity
* Presentation: The paper is reasonably well-organized, with clear explanations of the methodology and experimental setup.
* Figures and Tables: Visual aids like Figure 1 and Table 1 help illustrate the unlearning process and results.
* Writing: Some sections could benefit from clearer language and more precise descriptions, particularly in the methodological explanations.
## Originality
* Novel Approach: Introducing codebook features and SAEs for zero-shot unlearning in LLMs is a novel contribution.
* Advancement Over Prior Work: The method differs from traditional unlearning techniques that require retraining or specialized data structures.
## Significance
* Practical Impact: The ability to unlearn information without retraining has significant implications for data privacy and compliance with regulations like GDPR.
* Broader Implications: This work could influence future research on model editing and unlearning in complex models.
## Pros
* Innovative Method: The approach offers a new direction for zero-shot unlearning in LLMs.
* Efficiency: Avoids the computational cost of retraining or fine-tuning.
* Experimental Validation: Provides empirical evidence of the method's effectiveness.
## Cons
* Limited Scope of Experiments: The experiments focus on specific topics and a single language task (machine translation), which may limit the generalizability of the findings.
* Potential Overfitting to Specific Cases: The method might be tailored to the datasets used and may not perform as well on other types of data or tasks.
* Clarity in Methodology: Some methodological details are vague, making it difficult to fully assess the reproducibility of the approach.
* Evaluation Metrics: While BLEU and METEOR are standard, they may not fully capture the nuances of unlearning effects, especially in generative models.

---

### Official Review · Reviewer_57Qv · 2024-10-10
**New model training method that enables unlearning; insufficient evaluation**

**Rating:** 5
**Confidence:** 2

**Review:**

This paper proposes a machine unlearning method using codebook embeddings - a compressed, discrete representation of activations. This method requires the model to have been trained in a way that predictions depend on the cookbook features, which requires an additional loos term during model training.
In order to delete a topic, it is necessary to create a dataset with samples related to the topic that needs to be unlearned, replacing words indicative of this topic with words of other topics that remain appropriate in context. The goal is then to delete codes highly activated in the version of the dataset containing the topic and much less activated in the modified dataset.

The paper demonstrates some qualitative examples of this using a T5-small model on a Machine Translation dataset. The paper demonstrates how MT metrics on sentences containing a word change for 7 different words.

Overall, the problem the paper focuses on is important and useful for many downstream applications. The proposed method appears computationally simpler and conceptually more precise from existing methods. However, the paper is weak on experiments. While there is a demonstration of unlearning of a few words in the context of a single task, there is no comparison on the efficacy of unlearning with prior methods, and such a comparison would likely need to be done on a larger number of topics. A possible benchmark that may be relevant for this is TOFU: https://arxiv.org/pdf/2401.06121